# Chitosan as a Valuable Biomolecule from Seafood Industry Waste in the Design of Green Food Packaging

**DOI:** 10.3390/biom11111599

**Published:** 2021-10-28

**Authors:** Barbara E. Teixeira-Costa, Cristina T. Andrade

**Affiliations:** 1Programa de Pós-Graduação em Ciência de Alimentos, Instituto de Química, Universidade Federal do Rio de Janeiro, Avenida Moniz Aragão 360, Bloco 8G/CT2, Rio de Janeiro 21941-594, RJ, Brazil; ctandrade@iq.ufrj.br; 2Faculdade de Ciências Agrárias, Universidade Federal do Amazonas, Avenida General Rodrigo Otávio 6200, Manaus 69077-000, AM, Brazil

**Keywords:** chitin, chitosan, bioactive molecules, seafood industry residues, crustacean shells, green solvents, plasticizer, biodegradable food packaging, coatings, polysaccharides

## Abstract

Chitosan is a versatile biomolecule with a broad range of applications in food and pharmaceutical products. It can be obtained by the alkaline deacetylation of chitin. This biomolecule can be extracted using conventional or green methods from seafood industry residues, e.g., shrimp shells. Chitin has limited applications because of its low solubility in organic solvents. Chitosan is soluble in acidified solutions allowing its application in the food industry. Furthermore, biological properties, such as antioxidant, antimicrobial, as well as its biodegradability, biocompatibility and nontoxicity have contributed to its increasing application as active food packaging. Nevertheless, some physical and mechanical features have limited a broader range of applications of chitosan-based films. Green approaches may be used to address these limitations, leading to well-designed chitosan-based food packaging, by employing principles of a circular and sustainable economy. In this review, we summarize the properties of chitosan and present a novel green technology as an alternative to conventional chitin extraction and to design environmentally friendly food packaging based on chitosan.

## 1. Introduction

Natural biomolecules from renewable sources, such as chitosan, appear as good alternatives for the development of biodegradable food packaging. Chitosan is a linear polymer obtained by alkaline deacetylation of chitin. This biopolymer is extracted from crustacean exoskeleton, provided by seafood industry waste [1,2]. The biomass of crustaceans is comprised of proteins, calcium carbonate, chitin, pigments, and lipids [2]. The extraction process leads close to 20–30% of chitin, by applying mechanical and chemical treatments [2]. The alkali/acid conventional treatments of crustacean biomass to obtain chitin and chitosan can present low efficiency and may be considered disadvantageous when looking at the environment and labor conditions. However, green solvents, such as 1-allyl-3-methylimidazolium bromide and 1-ethyl-3-methyl-imidazolium acetate, can be used to replace those methods and improve the efficiency of chitin extraction and chemical modification processes [3].

The use of natural materials has advantages, in addition to reducing the consumption of petroleum-derived polymers and environmental pollution. These materials present good bioavailability, biocompatibility, and biodegradability. They also act as antioxidants, antimicrobials, with low toxicity and low allergenicity [4,5]. In the development of biobased films, their hydrophilic character should be considered, because humidity may be a problem to the shelf life of the product, as well as to the physical stability of the packaging. To overcome this issue, hydrophobic or less hydrophilic substances can be incorporated to the films modifying its physical properties. These substances, denominated plasticizers, are responsible for increasing the matrix free volume and molecular mobility of amorphous biopolymers, competing with intermolecular H-bonding [6,7]. Glycerol, sorbitol, polyethylene glycols (PEG), lipids and their derivatives are the most used plasticizers to biobased films, although novel sustainable alternatives, e.g., deep eutectic solvents (DES), have also been used [8,9,10]. These sustainable substances are obtained by complexation of a hydrogen bond donor (HBD) with a quaternary ammonium salt (hydrogen bond acceptor—HBA), producing liquids with physical and solvent properties analogue to ionic liquids (ILs) [11,12]. The mixture between choline chloride and glycerol (ChCl-Gly) has been the most used DES for plasticizing of chitosan-based coatings [8,9,13].

In this context, this review aims to summarize the application of waste from fishery industries as a source for chitin, mainly from crustaceans’ shells. As the deacetylated derivative chitosan is a bioactive molecule that can be used in several industrial processes, its biological properties and novel green technologies to design environmentally friendly food packaging are also reviewed.

## 2. Seafood Industry Waste

Seafoods are delicacies enjoyed globally because of their nutrients and sensorial properties, such as particular flavor and texture [14]. Fishes are an important source of proteins (~17% of protein intake), micronutrients, vitamins and essential fatty acids (omega 3) for human health [15]. The global production of seafoods has been growing rapidly during the last years, as a relevant component of the population’s diet and reaching more than 180 million metric tonnes in 2016 [1,16,17]. Of this, the capture method has been dominant and representing more than the half of this production [17].

Reports from FAO (Food and Agriculture Organization of the United Nations) have listed that the world production of crustacean in 2013 was 12,607.540 tonnes, of which Asia was the biggest producer (81.4%), followed by the Americas (13.7%), Europe (3.5%), Africa (1.1%) and Oceania (0.3%) [18]. In 2018, the global aquaculture production reached 82.1 million tonnes, representing a total sale of USD 263.6 billion, which were dominated by fish (54.3 million tonnes), molluscs and bivalves (17.7 million tonnes), and finally crustacean (9.4 million tonnes) productions [19]. From this report, Asia is by far the largest producer of all aquaculture species, reaching more than 72 million tonnes, followed by the Americas (4 million tonnes), Europe (3 million tonnes), Africa (2 million tonnes) and Oceania with around 200 tonnes. The major types of marine crustaceans globally produced by capture over the last two decades are presented in Figure 1 [19].

The intensive growing of the aquaculture industry, including crustaceans, fishes, molluscs, and others, is related to higher rates of fish consumption, which is among the most traded of food commodities [17]. However, a great number of challenges have shaken the seafood industry, mainly related to its environmental and social unsustainability. Some of these issues are due to the destruction of seabed when using bottom drag nets, overexploitation and illegal fishing, lack of sanitary conditions on fishing boats, killing untargeted sea animals that get trapped in fishing lines, disposition of inedible parts of sea animals directly into the water, pollution, forced and analogous slavery works, among others [1,14,17].

Fishing activities are known for generating a high amount of waste, since bones, shells, heads, skins, and visceral parts are not commonly eaten by humans. Between 2010 and 2014 the annual residue generated by global marine fisheries reached more than 9 Mt and close to 45% of it was from capture using bottom dragging nets [1]. The loss of global fishes was estimated to be close to 12 Mt per year, representing 10% of total seafood capture and aquaculture [20]. A recent report indicated that around 60% of all processed fishes are discarded, which are composed of ~20% muscles, ~18% viscera, ~15% bones, ~12% heads, 5% of scales and up to 3% of skin and fins [21]. However, the estimation of seafood losses and waste have been reported to be conflicting, because of the lack of information and uniformity in the assessment of data [15]. To these authors, waste generation is linked to issues during the different steps in the processing chain, such as primary and post-production, processing, distribution, and consumption, as well as a lack of uniformity in the definition of loss type, such as physical, quality, nutritional losses and others.

These seafood by-products end up being majorly utilized as animal feed, agricultural fertilizer, lipids extraction, biogas and biodiesel production [1,2]. Technological approaches of the valorization of seafood by-products are needed since those materials are great sources of bioactive compounds, such as amino acids and proteins, gelatin, collagen, vitamins, pigments, calcium carbonate, polyunsaturated fatty acids, and chitin [1,14].

## 3. Chitin and Chitosan

Chitin is an important structural component of the invertebrate exoskeleton of insects and other terrestrial arthropods (e.g., beetles and locusts), majorly present in crustaceans (e.g., crab, shrimp, and krill), which can be extracted from it or produced by microorganisms (e.g., fungi, and some algae) [22,23]. The earliest reports on chitin are from the early 19th century, when Antoine Odier extracted an alkaline-insoluble fraction from an insect’s exoskeleton and named it *Khiton*, which in Greek means tunic or an envelope [24].

Chitin is a linear polymer of mainly *β*-(1→4)-2-acetamido-2-deoxy-*D*-glucopyranose units and low amounts of *β*-(1→4)-2-amino-2-deoxy-*D*-glucopyranose residues (Figure 2) [23,25]. In a solid state, it can be found in three different crystalline polymorph structures, anti-parallel *N*-acetylglucosamine chains or α-chitin, parallel chains of *N*-acetylglucosamine or *β*-chitin and two parallel chains alternated with a single anti-parallel one, forming the *γ*-chitin [3,26]. The first two forms are the most frequent in nature and these arrangements may contribute to its mechanical properties [26].

About 20 to 30% of chitin, as well as proteins, minerals, and pigments, can be extracted from shrimp shell waste [27]. Researchers have investigated the transformation of crustaceans’ shells into a more valuable compound and a potential source of ingredients for the food industry [28]. Japan have been dominating the chitin/chitosan market since the 1980s until now, being responsible for more than 35% of its industrialization in 2015 and consuming more than 800 tonnes per year [29]. According to the Markets and Markets [30] the global market of chitosan was valued at around USD 476.6 million in 2016. Increasing projections indicate that chitosan market can reach around USD 1088.0 million in 2022, presenting a compound annual growth rate (CAGR) of 14.5% between 2017 and 2022 [30]. This study also estimated a market size of around 63.7 kilotons in 2022, from Asian countries, such as China and India, the leaders in the average propensity to consume. In another study, the chitosan market was estimated to increase to around 24.7% of CAGR till the year 2027, highlighting its prominent position to a wide range of industrial applications [31]. A substantial amount of waste is produced during seafood processing, making the extraction of chitin from this residue a good opportunity to add value to this market and to reduce the environment contamination.

However, chitin is an underutilized biomass resource, being mostly used for the chemical production of chitosan. About 33 kg of shrimp shells (in a wet basis), 0.02 L of diesel for bulldozer operation, 8 kg of HCl 32%, 1.3 kg of NaOH, 1.3 kWh of electricity and around 170 L of freshwater, are needed to produce one kilogram of chitin [32]. To obtain 1 kg of chitosan, 1.4 kg of chitin, 5.18 kg of NaOH, around 1.1 kWh of energy and 250 L of water are needed [32]. Although during the production of chitin and chitosan, some chemicals and other resources are used, the use of seafood shells as raw materials helps prevent the accumulation of composting materials that end up polluting the environment, and could alter the sea life cycle, and save composting emissions such as ammonia [32]. As chitin is a highly acetylated polymer, insoluble in water, traditional methods, such as chemical deacetylation, or enzymatic hydrolysis, are most commonly used to make it less hydrophobic, which results in the production of chitosan [25,26,33,34]. Its chemical deacetylation involves an alkaline treatment at high temperatures. Different factors, such as alkali concentration, processing time, chitin:alkali ratio, temperature, and chitin source, can affect the degree of *N*-acetylation (DA) of chitosan.

To overcome the use of hazardous chemicals during chitin extraction, an alternative green extraction method has been reported [35]. Some researchers have easily used ionic liquids, such as 1-allyl-3-methylimidazolium bromide (AMIMBr), followed by demineralization by employing 1.5% of citric acid and subjecting it to different extraction temperatures (80–120 °C) for 6–48 h, achieving a chitosan yield up to 12.6% [36]. Moreover, according to some authors, 1-ethyl-3-methyl-imidazolium acetate was able to recover higher amounts, around 94%, of chitin from shrimp shells [37]. To these authors, the extracted chitin exhibited a higher M_w_ and the processing conditions were less harsh than the conventional methods to chitin extraction. This result demonstrates the benefits of using green solvents to improve chitin extraction.

Chitin is insoluble in many organic and inorganic solvents, due to its high intra and intermolecular hydrogen bonds. Its dissolution is essential to estimate the molecular weight (M_w_). However, the presence of aggregates in chitin dispersion trickles the measurements by light scattering, which fails in determining molecular weight [38]. To bypass this, green solvents as ionic liquids and deep eutectic solvents, e.g., 1-allyl-3-methylimidazolium bromide, 1-ethyl-3-methylimidazolium alkanoates and others, have been recently used for chitin solubilization [3]. A schematic summary of conventional and green extraction techniques and potential applications of chitin/chitosan are presented in Figure 3.

Chitosan is a natural polysaccharide derived from chitin. Both polysaccharides are the most abundant biopolymers from an animal source [22]. The first work with chitosan dates from the 1850s, when Charles Rouget reported that he had obtained a novel *chitine modifiée* after treating chitin with a concentrated alkaline solution under reflux, making modified chitin soluble in organic acids [24]. Many years later, in the 1890s, shells from crabs, scorpions and spiders were treated by Felix Hoppe-Seyler with a solution of potassium hydroxide at 180 °C. This researcher obtained a soluble acid solution product named chitosan [22,24]. Up to the 1950s, chitin and chitosan gained considerable attention due to the increasing exploitation of natural polymers.

Chitosan is a versatile natural polymer with a broad range of applications in food products, mainly because of its biodegradability, biocompatibility, nontoxicity, and bioactive properties, such as antioxidant, antimicrobial, and anticancer activities [4]. Chitosan is a linear random copolymer of *N*-acetyl-*D*-glucosamine (*A*-units) and deacetylated *D*-glucosamine (*D*-units) units connected by *β*-1,4 glycosidic linkages (Figure 4), obtained from the chemical alkaline deacetylation of chitin [4,33]. In each repeating unit, the chitosan molecule has three functional groups, i.e., primary, and secondary hydroxyl groups and amine groups. These groups allow chemical changes under pronation, and influence its biological, mechanical, and physical-chemical characteristics, including solubility, hydrophilicity, and crystallinity [3,4]. The amine groups, NH_2_, located on C-2 of the rings on the *D*-glucosamine repeated units, may be pronated and induce changes on chitosan chains under acidic conditions, acquiring a polycationic characteristic [4,28].

The properties of chitosan in solution depend on the degree of acetylation (DA), chain length, and distribution of acetyl groups along the chain [26]. The DA is given by the ratio between its acetylated and deacetylated units and is expressed as molar percentage (mol%). The DA can range between 0.05 and 0.30. Below DA = 50%, chitosan is soluble in aqueous acidic media [25,33]. The chitosan origin has an influence on its DA as well as on its M_w_. Generally, microbial chitosan exhibited a lower DA than those extracted from the exoskeleton of crustaceans [34]. Heterogenous deacetylation played on solid-state chitin may produce a block-wise distribution of acetyl groups causing chain linkage and consequently affecting its dissolution and M_w_ quantification [38]. The DA and the M_w_ are the most important factors contributing to the physicochemical identity of chitosan and influence its structure-properties relationships, e.g., solubility, viscosity, and gelling abilities, among others [39,40].

The main characterization of chitosan starts with the determination of its molecular weight in solution, followed by the DA quantification and finally the distribution of acetyl groups along the backbone chain (by nuclear magnetic resonance—^13^C NMR) [38]. Various methods have been used to quantify the molecular weight of chitosan, e.g., viscosimetry, light scattering, osmometry, size exclusion chromatography and more recently the multi-detection asymmetric flow-field flow fractionation [39]. The chitosan DA can be determined by infrared spectroscopy, potentiometric titration, and elementary analysis, although, ^1^H liquid state and solid-state ^13^C NMR spectroscopy have been favored [38,40]. Besides these parameters, the intrinsic viscosity, [η], is also an important factor to be considered for chitosan characterization. The [η] may be determined by double extrapolation to zero concentration of Huggins’ and Kraemer’s equations (Equations (1) and (2)) [28].
(1)$$\frac{\eta_{sp}}{C}=\;\left[\eta \right]\;+\;k^{\prime }{\left[\eta \right]}^{2}C$$
(2)$$\frac{\left(Ln\;\eta_{rel}\right)}{C}=\;\left[\eta \right]\;+\;k^{\prime \prime }{\left[\eta \right]}^{2}C$$
where η*_rel_* and η*_sp_* are the relative and specific viscosities, *k*′ and *k*″ are the Huggins’ and Kraemer’s coefficients, respectively, and *C* is the chitosan solution concentration.

The intrinsic viscosity is essential to determine the viscosity average molecular weight, *M_v_*, using the Mark-Houwink-Sakurada relationship as expressed in Equation (3) [28,38].
(3)$$\left[\eta \right]\;\left(mL\;g^{-1}\right)\;=\;K{\bar{M}}_{v}^{a}$$
where [η] is the intrinsic viscosity, *K* is the intercept and *a* is the equation slope, and $\bar{M}$ is the viscosity average molecular weight for chitosan.

Brugnerotto et al. [41] characterized chitosan by steric exclusion chromatography with different DA (0.5 to 25 mol%) and found values for the *K* and *a* constants varying from 0.080 to 0.068, and 0.796 to 0.800 (random coil), respectively, when the used solvent was AcOH 0.3 M/AcONa 0.2 M at 25 °C. High values (>0.800) of *a* coefficient indicates a semi-rigid feature of the polysaccharide with a very stiff chain, and when *a* presents a low value (<0.650) it is an indication that the chain has adopted a compact sphere conformation. This variation affects their hydrodynamic volume, dimensions, and viscometrical properties [38]. However, the most frequent values reported for the constant *a* ranged between 0.7 and 1.0, indicating that chitosan behaves like a flexible or linear conformation depending on the solvent [42].

Chitosan is a versatile material, used for a wide range of applications. It is considered an eco-friendly biopolymer since its biodegradability in various environments as other natural polysaccharides (Figure 5). For environmental purposes, it can be used for water purification, as filtration membrane, flocculation/coagulation of dyes and proteins, and wastewater treatments, as it is a competent sorbent material for heavy metals, pesticides, and dyes [26,29,33,43]. In agricultural production, chitosan (CS) can be applied as micro/nanoparticles acting as fertilizer carriers aiming to mitigate deleterious effects to plants and human health [33,44]. It can be used in the development of electrochemical sensors in combination with conducting materials, ionic liquids, green solvents, and carbon nanotubes, and many other purposes in the chemistry industry [29,43].

The most important application of chitosan is oriented to pharmaceutical, cosmetic, nutrition and food markets, mainly because of its classification as a safe substance (GRAS—Generally Recognized as Safe), by regulatory agencies of many countries. In the United States, CS is classified as a preservative and antioxidant substance by the Environmental Protection Agency [45]. In Brazil, CS is approved as a functional health ingredient for food products by the National Health Surveillance Agency of Brazil, Agência Nacional de Vigilância Sanitária [46]. In the food and nutrition field, CS has a significant market in Asia, mainly Japan, China, and South Korea, followed by the USA in North America, and Europe, who reached more than 2000 metric tonnes in 2018 [29]. This demand is mostly related to the utilization of CS as a nutraceutical/functional ingredient in dietary supplements and as a natural additive with a wide range of functions in food products.

The countless biological activities of chitosan, such as antioxidant, antimicrobial, antifungal, anti-inflammatory, wound healing, immunostimulant, and prebiotic benefits have been well documented by different studies [29,33,34,47]. Chitosan is non-digestible in the upper gastrointestinal tract, influencing/blocking the absorption of carbohydrates, fats, and cholesterols, and contributing to a faster intestinal transit time. Additionally, it is reported that CS could increase the excretion of atherogenic saturated fatty acids, which could lead to a reducing risk of developing a chronic disease [34]. Furthermore, as a dietary fiber, chitosan can be considered as a prebiotic material, due to the improvement of gut microbiota [47]. For this reason, CS has been commonly used as a dietary product as part of an effective body weight regimen and control of fat mass, even though this effect is not completely unanimous in the literature [34,47].

A recent single-blinded, placebo controlled and random clinical study (500 mg of CS) for body weight reduction found that CS from fungal origin reduced the average body weight by up to 3 kg after 90 days. Additionally, it promoted an improvement in body composition, and anthropometric parameters [48]. In another work that performed a systematic review of CS consumption, through a meta-analysis of 15 randomized controlled trials, found a significant reduction (*p* < 0.05) in body weight and body fat of overweight/obese participants that received CS supplementation [49]. However, studies aiming to investigate this question may differ, as there are different sources and types of chitosan. Moreover, approaches using different dosage, timing, methodologies to assess end points (weight loss vs. fat loss), lack of well-constructed clinical trials, and other issues can affect these studies [50].

Another common utilization of CS by the food industry can be highlighted due to its excellent antimicrobial/antioxidant properties. To this purpose, CS can be used in many different forms and applications, e.g., as a solution, in powders, gels, beads/particles, fillers, films, casting or sprayed coatings, as well as its blends. The use of chitosan for preservative purposes of foods as an antimicrobial/antifungal agent has been well reported by many authors [22,51,52,53,54]. CS has a broad range of antimicrobial activity against pathogenic microorganisms, a bacteriostatic effect against Gram-positive and Gram-negative bacteria, e.g., *Escherichia coli*, *Vibrio cholerae* and *Shigella dysenteriae*. CS also acts against yeasts and molds, generally being more active against them than with bacteria [22,47,55]. A summary of chitosan applications is listed in Table 1.

Mainly, the physical state as well as the solubility of chitosan are the most significant factors that affect its antimicrobial activity. It was observed that microbial inhibition by chitosan only occurred in acidic mediums, where it is soluble and has a net positive charge [53]. Although not completely elucidated, antimicrobial, antifungal, and antiviral properties have been correlated to interaction with chitosan protonated groups, involving electrostatic stacking at the cell surface, chelation of essential trace elements and metalloenzymes, protonation forms and cell membranes disruption, and blockage of RNA transcription from DNA when chitosan molecules are inside cell’s nucleus [22,25,34,51,56]. This electrostatic staking occurs by the interaction between protonated amino groups of glucosamine and negative cell membrane of microorganisms, affecting its integrity and permeability and consequently altering their metabolism, which can lead to cell death [22]. Moreover, CS can prevent the production of toxins and bind essential trace metals and spores acting as a chelating agent, which affects the essential supply of nutrients, inhibiting microbial growth [22].

Other environment factors, such as molecular weight, DA, positive charge density, pH, ionic strength, and temperature, can also influence its antimicrobial activities [53,55]. Furthermore, chitosan easily forms quaternary ammonium salts at low pH values. Thus, organic acids, such as acetic, formic, and lactic acids can dissolve it [27]. Acetic acid, generally at 1% concentration, is the most common organic acid used for chitosan dissolution [57]. When chitosan is dissolved in an acidic media, generally below pH 6.5, its amino groups in the main chain protonate and turns in to a cationic polysaccharide. These positively charged groups, protonated amines, allow the interaction with a wide range of negative charged molecules forming complexes. Some of these negative charged substances are anionic synthetic or natural polymers, dyes, lipids, fats and cholesterol, metals ions, enzymes, biological cells, DNA and RNA [3].

### 3.1. Food Packaging

Packaging protects foods from the environment. Over the years, packages have changed to follow the different demands of consumers and the evolution of the food industry. Packaging is responsible for maintaining food quality, improving safety and shelf life, as well as providing label information of ingredients, and promoting the worldwide distribution of food products to reach final consumers [4,77]. From all of these, the maintenance of food quality across the whole supply chain is the most critical factor when choosing a packaging material [22,78]. To fulfill these functions, the materials for the packaging design must have good thermo-mechanical properties, to be properly processed by different means, resist the surrounding aspects through distribution, exhibit an adequate shelf life for product storage and prevent cross contamination of unwanted substances to their content [79]. Moreover, packaging technology is dynamic and has evolved since ancient times until nowadays changing humans’ lifestyle. Through this timeline, different materials have been used for food packaging, such as glass, metal, paper, and plastic, though these last two are more often utilized for this purpose [4,57,80].

Conventional plastic packaging made from synthetic polymers are items of one-time use, generally discarded in the environment after using. Around 40% of petrochemical-based plastics are used for packaging purposes and close to 60% of plastic packaging are used to pack food and beverages [81]. About 95% of plastic packaging are wasted after a single-use cycle and end up in the environment, where it can be broken down into micro or nanoplastics [82,83]. These microplastics are carried out by ocean currents and can be ingested by the sea-wild life and birds, in which they accumulate in their digestive system, modify their health, and end up in the food chain [84].

Another issue that negatively affects plastic packaging production is concerns about human occupational health, since different hazardous chemicals added intentionally and unintentionally are used during processing [81]. Moreover, during the manufacturing of synthetic petroleum-based plastics, greenhouse gases and particulates are released in the environment, also contributing to world pollution [79]. In addition to all of this, packaging manufacturers face a new market challenge due to consumer demand for biobased resources for food packaging [79,85]. Thus, all these circumstances have been stimulating the search for new biobased materials and the development of biodegradable packaging from renewable sources as strategies to mitigate pollution and sustain the planet. In response to these challenges, there is a great opportunity for the development of novel food packaging. In this context, biobased, biodegradable, compostable, edible, active and intelligent packaging are innovative trends in the field.

#### 3.1.1. Chitosan-Based Films to Food Packaging

Chitosan is among the most studied polysaccharides for the development of film/coating packaging. This is justified by its versatility, good physical-chemical and biological properties. Mechanical properties are affected by the chitosan DA, solvent, pH, concentration, viscosity, and molecular weight [4]. Moreover, chitosan is generally less expensive and commercially available when compared to other biopolymers [25]. The first reports of chitosan films are from the mid-1930s, when George W. Rigby patented it [24]. Since then, the employment of chitosan as films, coatings, or composites, to different products has been extensive. Recent works on the sustainable application of chitosan-based films for food packaging were reviewed by Haghighi et al. [22] and Mujtaba et al. [4].

As with any other packaging, edible films and coatings must protect the food integrity, maintaining quality, controlling mass and volatile losses, extending its shelf life, preventing surface contamination, offering mechanical protection, and improving its sensory properties (Figure 6) [22,86,87,88]. Although, it is not expected that edible films can completely replace all conventional packaging, they can be used to significantly reduce the use of petroleum-based plastics, decrease food losses, and reduce the environment pollution in the long-term. Edible films and coatings may have in their composition two main based materials, a biomacromolecule-based matrix, and additives, such as plasticizers, cross-linkers, other reinforcements substances and functional ingredients [88]. These materials can be used alone or blended. As edible packaging is composed of natural-based materials, it can be classified as a subgroup of biobased and biodegradable packaging [87]. Generally, polysaccharides, proteins, lipids, and their mixtures are used for the development of edible films and coatings [88].

Films based on chitosan are environmental-friendly materials and can be degraded by microorganisms [89]. Leceta et al. [90] have studied the environmental assessment between two different food packaging systems, polypropylene, and chitosan films, during different life cycle stages. For them, from about 1 kg of chitosan-based film waste, 20% can be considered as compost and the remaining 80% degraded as organic matter by microorganisms. Furthermore, these authors found that the highest positive impact of chitosan-based films was related to the end-of-life scenarios, especially associated with composting and carcinogens, when compared to polypropylene films, which can provide a reduction of environmental pollution generated by the food packaging industry.

In another work, the biodegradability of chitosan-based films on three different soils, industrial compost, commercial garden soil, and soil from a vineyard, was evaluated. The authors found that the properties of the chitosan-based film deteriorated in less than 3 days and its biodegradation occurred in all tested soils after 14 days [91]. Moreover, when these films were incorporated with Quercus polyphenol extract its active properties in compost and garden soil occurred in 6 days, whereas the total biodegradation process was not completed in the vineyard soil during the 14 days. These authors also found that adding water to the soil decreased the rate of biodegradation from the chitosan film on the terrestrial environment. These findings are relevant to confirm the biodegradability of chitosan films when using it as a sustainable alternative food packaging.

Pure chitosan films can only be prepared via solvent casting. In this process, chitosan is dissolved using enough solvent, mainly, acidified water, placed on a flat surface and let to dry until constant weight. Generally, these pure films are reported to exhibit a smooth, continuous, and compact surface, but are brittle and fragile possessing low mechanical properties [26,33,92]. To overcome the low mechanical performance and high sensitivity to water, the addition of plasticizers and other different approaches have been indicated. Some of these use crosslinking, complexation, graft copolymerization, surface coating, filler incorporation and others [22]. Blending chitosan with other polymers by solution or extrusion blending to form composite films could also be a strategy. Binary or ternary blends, e.g., CS-gelatin, CS-alginate, CS-gelatin-caprolactone, have been developed for food and biomedical applications [33]. The formation of a cohesive film-forming material resistant to rupture between CS and cellulose in the paper industry have also been reported, as the wet strength of paper can be improved by CS addition [33]. In another study, CS was used as a surface coating agent on printing paper, improving its mechanical properties, such as its strength, and inhibited the growth of bacteria, such as *E. coli* [93].

Regarding the mechanical properties, tensile strength (TS) and elongation at break (EB) are two of the most important characteristics for film applications [25]. TS is related to the maximum tensile stress that a film can hold up [94]. Moreover, the TS of chitosan films with high and low molecular weight was described as dependent on storage time due to the conformational changes of chitosan molecules by the free volume reduction [95]. The type of solvent acid used to prepare chitosan-starch composite films also influenced TS according to the following order: lactic acid < malic < acetic acid [96]. In another work, chitosan film displayed a significantly lower TS, 18.252 MPa, but a higher EB, ~40%, than gelatin and composite films [97]. In the development of chitosan-zein composite films, Sun et al. [94] found that TS was influenced by different plasticizers, sorbitol < glycerol and PEG-400; this result was explained by the number of hydroxyl groups in the plasticizer. Sorbitol has the capacity of binding a higher number of water molecules. The TS values may also decrease with the increase in pH because of a lower chitosan dissociation [57]. Furthermore, in chitosan composite films with quinoa protein, it was observed that TS and EB were influenced by the presence of protein in the films [98]. Some application examples of chitosan-based films are listed in Table 2.

Water vapor and the oxygen barrier are relevant factors to consider in the development of food packaging since both can influence the food quality and shelf life. The evaluation of water vapor permeability (WVP) of polysaccharide-based films provides information of diffusion and solubility of water molecules through the polymeric matrix and will be influenced by its composition. In this context, the WVP of high and low molecular weight chitosan films was investigated by Kerch and Korkhov [95]. They found that the high molecular weight chitosan films exhibited higher permeability. The addition of a natural phenolic antioxidant, protocatechuic acid, significantly decreased the WVP of chitosan composite films [112]. Furthermore, the addition of vegetable olive and corn oils displayed a significant effect decreasing the WVP by reducing the hydrophilic content of the film [103]. Different mechanical and barrier properties of chitosan films have been reported. These properties are affected by chitosan molecular weight, solvent, plasticizer content, film composition, and pH, which makes the comparison between them more challenging.

#### 3.1.2. Plasticizing Biodegradable Films with Green Solvents

In the development of biobased films, the hydrophilic character of their natural macromolecules matrix should be considered since water molecules may be a problem for the shelf life of the product as well as the physical stability of the packaging. To overcome this issue, hydrophobic or less hydrophilic substances can be incorporated to the films modifying the physical properties. These substances are denominated plasticizers and are generally of low molecular weight. Plasticizers are responsible for increasing the matrix free volume and molecular mobility to amorphous biopolymers, competing chain-to-chain H-bonding along the polymer chains [6,7]. They are often used to reduce the brittleness by reducing the polymer interchain interactions by putting themselves between polymer molecules, although they do not chemically bind to it [88].

The most commonly used plasticizers are glycerol, sorbitol, polyethylene glycols (PEG), lipids and their derivatives, such as fatty acids, phospholipids, lecithin, oils, and waxes, low molecular weight sugars, e.g., fructose-glucose syrups and honey, and others [6,88,92]. Crosslinkers and micro/nano-reinforcements can be incorporated into film packaging aiming superior tensile, gas and water barrier properties. The improvement of coatings cohesive strength can be achieved by application of other physical treatments, i.e., irradiation, by crosslinking formation [6]. More recently, films’ plasticization can be performed using deep eutectic solvents (DES), as sustainable solvent systems [8,9,10].

DESs are obtained by the complexation of a hydrogen bond donor (HBD) with a quaternary ammonium salt (hydrogen bond acceptor—HBA), producing liquids with physical and solvent properties analogue to ionic liquids (ILs) [11,12]. Ionic liquids are liquid solvents with a melting point below 100 °C and are formed from systems composed primarily of one type of discrete anion and cation. Initially, DESs were extensively used to decrease the temperature of molten salt. Nowadays, they have been used for a wide range of applications, from lubrication of steel to synthesizing drugs [12,113].

DESs contain large asymmetric ions of the HBA that have low lattice energy and so, low melting points, caused by hydrogen bonding formation. A wide range of mixtures of substances can form DES, which was critically reviewed by Smith et al. [12]. For plasticizing film packaging the most common DES used are the mixture of choline chloride-citric acid (ChCl-CA), choline chloride-urea (ChCl-Urea) and choline chloride-glycerol (ChCl-Gly). The latter has been used mainly for chitosan-based coatings [8,9,13]. Particularly, the eutectic mixtures of ChCl:Gly at a molar ratio of 1:2 were found to efficiently plasticize starch and starch/zein blends, and chitosan and chitosan/micro-crystalline cellulose/curcumin composites [9,10].

Many advantages can be pointed out from using DES as they are much cheaper, safer and easier to manufacture than ionic liquids. Moreover, DES are chemically and thermally stable, non-flammable, possessing high dissolution ability, lack of flammability, low volatility, low melting point and tailorability, as well as being water neutral, and low- or non-toxic. The DES toxicity, including cytotoxicity and phytotoxicity, is highly dependent on the testing species used on the mixture and their responses, as well as their physicochemical properties, concentration, and salts counteranion species [13]. In the work of Mbous et al. [114], the toxicity of some DES was tested, and they found that pure ChCl exhibited lower cytotoxic values than its aqueous solutions (EC50 30 ≥ ChClaq ≤ 40 mM), suggesting that ChCl did not dissociate after crossing the cellular membrane, implying a nontoxic profile of ChCl-based DES, although more in vivo or in vitro studies could be further investigated [114]. As they are often biodegradable, they can be considered green solvents. DESs are appropriate for biobased packaging processing and application. For the preparation of polysaccharide films, the DES mixture should be separately prepared and incorporated to the matrix materials or can be formed in situ after heat homogenization with other components [13].

## 4. Conclusions

In this review, we discussed the increasing industrial seafood production and the challenges related to its growth, mainly due to the high waste generation and the consequences of the environmental pollution. Valuable biomolecules for the food and pharmaceutical industries, such as chitin and chitosan, can be extracted from these residues. Eastern markets have been leading the industrialization of these biopolymers. Japan may be cited as an example for consuming more than 800 tonnes per year of both biopolymers. Chitin and chitosan are linear polysaccharides. The first is insoluble in many organic and inorganic solvents. This limitation is overcome by alkaline deacetylation forming chitosan, which is soluble in acid solutions. Chitosan is a versatile biomolecule with a broad range of applications in food products, mainly because of its biodegradability, biocompatibility, nontoxicity, and bioactive properties. These biological properties as well as their mechanical and physicochemical characteristics depend on the chitosan acetylation degree, chain length, and distribution of acetyl groups along the chains. Because of those versatile properties, chitosan is among the most studied polysaccharides for the development of film/coating packaging. The first reports on chitosan films are from the mid-1930s and until now it has been applied differently in the development of biodegradable food packaging. One of the main challenges for using chitosan to design food packaging is related to its physical properties, e.g., water vapor permeability and processability. The use of green reagents as DESs, are allies to defeat those difficulties and to improve the mechanical properties of chitosan films. The most commonly used DES to plasticize chitosan-based films is choline chloride-glycerol. Many advantages can be pointed out by using green solvents as they can be safer, thermally stable, non-flammable, and easier to manufacture than other conventional reagents and even ionic liquids. Moreover, DESs can exhibit low toxicity and are considered biodegradable and nonpolluting, making them a valuable additive to the development of chitosan-based films. Furthermore, the exploitation of chitin and chitosan from seafood waste contributes to the seafood chain production by reducing the amount of crustacean residues discarded in the environment. Indeed, this is an important approach in a circular and sustainable economy. From another perspective, the conventional extraction of both biomolecules requires the use of hazardous chemicals, such as sodium hydroxide and hydrochloric acid. However, innovative technological approaches, like the use of green solvents, can once again be used to replace those, making the extraction process more efficient and more environmentally friendly.

## Figures and Tables

**Figure 1 biomolecules-11-01599-f001:**
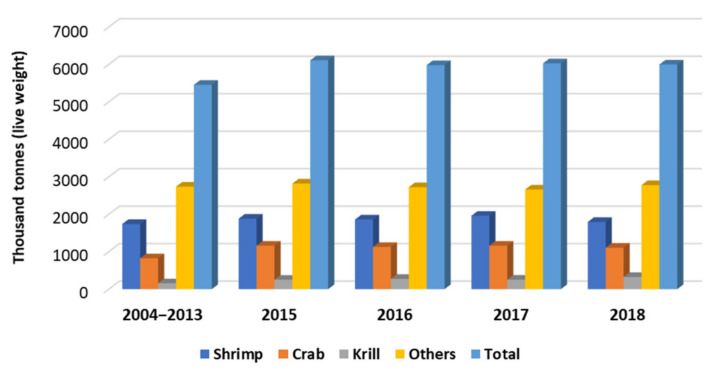
Major types of marine crustaceans globally produced by capture over the last two decades.

**Figure 2 biomolecules-11-01599-f002:**
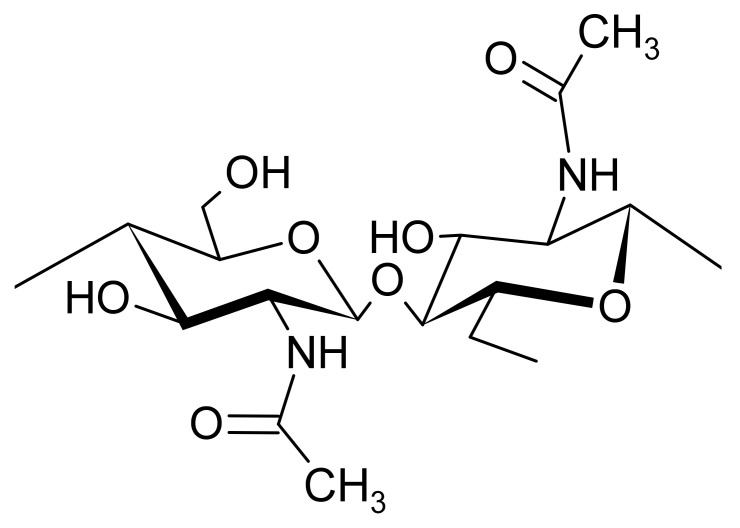
Chemical structure of chitin.

**Figure 3 biomolecules-11-01599-f003:**
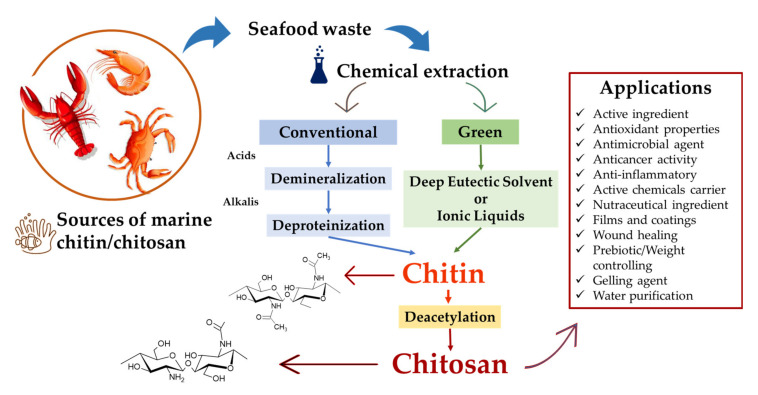
Summary of chitin/chitosan extractions and applications.

**Figure 4 biomolecules-11-01599-f004:**
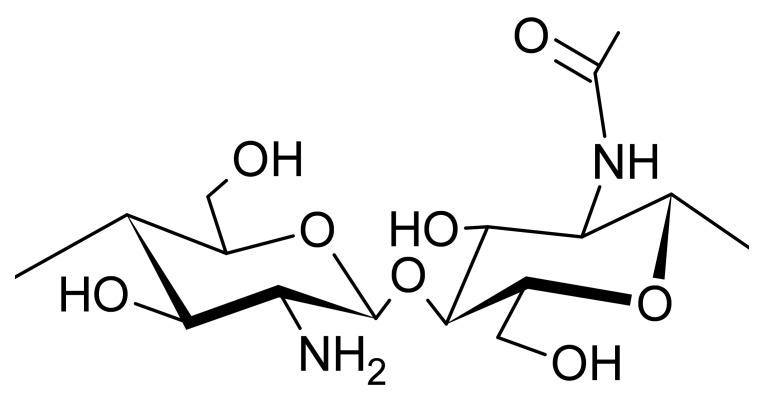
Chemical structure of chitosan.

**Figure 5 biomolecules-11-01599-f005:**
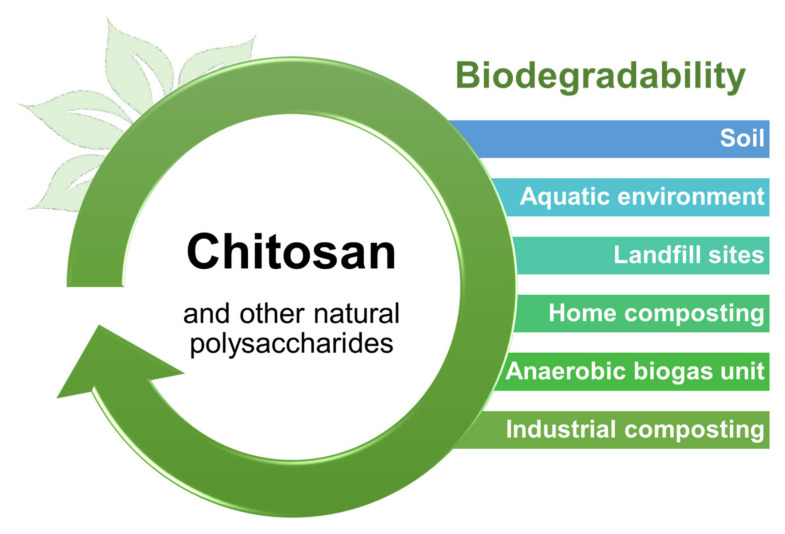
Biodegradability of chitosan and other natural polymers on various environments.

**Figure 6 biomolecules-11-01599-f006:**
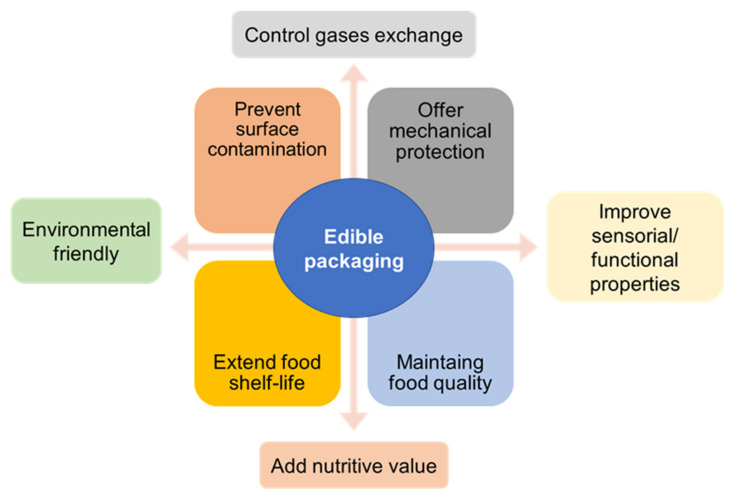
Main purposes of an edible food packaging.

**Table 1 biomolecules-11-01599-t001:** Broad application of chitosan for different purposes.

Application	Purpose	Product	Reference
Nutraceutical/functional ingredient	Body weight reduction	Oral capsules	[50]
	Food packaging	Assai polyelectrolyte complexes	[58]
Additive in food products	Texture controlling agent	Rice noodles; meat product	[59,60]
	Emulsifying/gelling agent	Films; microparticles	[61,62]
		Pastes	[33]
Biological, antimicrobial or antioxidant activity	Preservation agent in powder	Rice	[63]
	Film/coatings	Fish fillet	[64]
		Chicken fillet	[65]
		Meat cutlets	[66]
		Fruits	[57,67,68]
	Nanofibers	Active food packaging	[69]
	Hydrogels	Delivery of enzymes	[70]
	Liposomes	Delivery of bioactive substances	[71]
Dentistry application	Hydrogel	Remineralization of enamel surface	[72]
Biomedical application	Layer-by-layer coating	3D multilayered microchannels in hydrogels	[73]
	Nanoparticles	Chitosan functionalized magnetic nanoparticles	[74]
	Self-assembly system	Polyelectrolyte complexes	[75]
	Nanovesicles	Chitosan/nutriose coated niosomes	[76]

**Table 2 biomolecules-11-01599-t002:** Broad application of chitosan-based films and their main results.

Films Matrix	Composite Ingredients	Applications	Main Results	References
Chitosan/Zein	Glycerol, PEG-400, and sorbitol	Potential food application	Permeability increased with higher plasticizer concentration. PEG-400 promoted a better barrier property.	[94]
Chitosan	Aloe vera gel and silver nanoparticles (SNPs)	Potential biomedical applications	SNPs decrease the crystallinity of the films. SNPs affected the structure and viscoelastic behavior of chitosan films.	[99]
Chitosan	Glycerol	Strawberries	Protection of the fruit against fungi. Maintenance of flavor, appearance, texture, and aroma.	[100]
Chitosan/purple yam starch	Glycerol	Coating of apples	The films preserved the quality of apples for four weeks of storage. Weight loss from the coated apples was less significant than the uncoated.	[101]
Chitosan	Apple peel polyphenols (APP)	Potential bio-composite food packaging material	APP significantly increased thickness, density, solubility, opacity, and swelling ratio of films. APP decreased the moisture content, water vapor permeability, tensile strength and elongation at break of the films.	[102]
Chitosan/organoclay nanocomposites (OrgMMT)	Olive oil and corn oil as plasticizers	Potential food packaging applications	OrgMMT significantly reduced the elongation at break of all oil containing samples, acting as stress concentrator upon deformation. Corn oil was a less effective as plasticizer than olive oil.	[103]
Chitosan/gelatin	Red grape seed extract and *Ziziphora clinopodioides* essential oil	Potential food packaging applications	The addition of red grape seed extract and *Z. clinopodioides* essential oil improved total phenolic content, antibacterial and antioxidant activities, thickness, and water vapor barrier property.	[104]
Chitosan/corn starch	Glycerol as plasticizer	Potential food/pharmaceutical applications	The water vapor permeability and moisture content of films increased with an increase in chitosan concentration.	[105]
Chitosan hydrochloride (CHC)	Glycerol as plasticizer and epigallocatechin gallate (EGCG) nanocapsules (NCs)	Potential food packaging applications	The incorporation of nanocapsules into the CHC films increased their tensile strength and percentage of elongation at break. The produced films could prevent lipids oxidation of fatty food products	[89]
Chitosan	Propolis extract	Potential food packaging applications	All chitosan/Propolis films inhibited bacteria on contact surface underneath the film. Mechanical properties, oxygen and moisture barrier, antioxidant and antimicrobial properties were improved by the addition of Propolis extract into the films.	[106]
Chitosan	Clove oil (*Syzygium aromaticum*)	Cooked pork sausages	The shelf life of cooked pork sausages increased from 14 to 20 days with the chitosan/clove oil films.	[107]
Chitosan/gelatin	Glycerol as plasticizer	Beef steaks	Myoglobin oxidation during retail display was reduced and the percentage of deoxymyoglobin increased with gelatin content in films.	[108]
Chitosan	Glycerine, chokeberry pomace extract	Potential food packaging applications	Enhanced water vapor permeability and reduced oxygen permeability by addition of chokeberry extracts. The films showed significant antioxidant properties.	[109]
Chitosan	*Urtica dioica* leaf extract derived copper oxide (CuO) and zinc oxide (ZnO) nanoparticles	Food film packaging and shelf life of guava fruit	Low antioxidant activity and greater antimicrobial properties. Decreased moisture content, water holding capacity, and solubility. The nanocomposite chitosan films improved the quality and shelf-life attributes of guava by one week when compared to unpackaged fruits.	[110]
Multilayer chitosan composites films	Sodium sulfoethyl cellulose (SEC), sodium alginate (ALG), and sodium hyaluronate (HA)	Potential biomedical and food applications	Effect of the polyelectrolyte complex layer on the properties of the poly-layer composites decreases in the order CS/SEC > CS/HA > CS/ALG.	[111]

## Data Availability

Not applicable.

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
