# Peer review of "Chitosan as a Valuable Biomolecule from Seafood Industry Waste in the Design of Green Food Packaging"

_biomolecules, 2021, doi:10.3390/biom11111599_

Round 1

Reviewer 1 Report

The paper topic is according with the actual research trends to find alternative and green solutions to preserve the food quality. Comments: 1. The review objectives and paper aim are missing from the introduction section 2. In the abstract it is mentioned that the review …….”present novel green technologies as alter- natives to conventional chitin extraction…” but in the content of manuscript are presented only green alternatives regarding the using of plasticizers based on DES. 3. The paragraph from lines 55-59 is similar with the paragraph from lines 430-435. 4. Generally chitosan has good film forming properties without plasticizers. This biopolymer is used in polyelectrolyte complexes with other polysaccharides (i.e. hemicelluloses) to improve the film flexibility and stability. 5. The section 3.1 and 3.1.1 contains limited information about the antimicrobial properties of chitosan biopolymer and other application of the chitosan in food packaging (i.e. coatings for paper/board, composite films with mineral nanoparticles – ZnO…etc) knowing the fact that the in these type of packaging chitosan is used as biopolymer matrix to embed other compounds with active and functional properties. 6. In the conclusion section it is mentioned also, ..”From another perspective, the conventional extraction of both biomolecules requires the use of hazard chemicals such as sodium hydroxide and hydrochloric acid. However, innovative technological approaches, as the use of green solvents, can once again be used to replace hazard chemicals, making the extraction process more efficient and more environmentally friendly…” the using of green solvent to extract the chitin from seafood waste, but in the manuscript is not described these extraction methods…

Author Response

Dear reviewer,

Many thanks for your comments. We have incorporated all your suggestions and made the appropriate corrections. The authors appreciated your careful evaluation and comments. We have incorporated two figures in the text to improve the manuscript comprehension. All changes made were written in red in the revised manuscript.

Reviewer 1 comments:

The paper topic is according with the actual research trends to find alternative and green solutions to preserve the food quality.

  1. The review objectives and paper aim are missing from the introduction section

Response 1: The aims of this manuscript were inserted in the Introduction section (Lines 56-59).

  1. In the abstract it is mentioned that the review …….” present novel green technologies as alter- natives to conventional chitin extraction

…” but in the content of manuscript are presented only green alternatives regarding the using of plasticizers based on DES.

Response 2: Green strategies to chitin extraction are presented on Lines 138-150. The use of ionic liquids is an example of a green methodology to extract chitin from crustacean’s shells.

  1. The paragraph from lines 55-59 is similar with the paragraph from lines 430-435.

Response 3: The paragraph on Lines 49-55 was amended.

  1. Generally chitosan has good film forming properties without plasticizers. This biopolymer is used in polyelectrolyte complexes with other polysaccharides (i.e. hemicelluloses) to improve the film flexibility and stability.

Response 4: The information was complemented (lines 341-346).

  1. The section 3.1 and 3.1.1 contains limited information about the antimicrobial properties of chitosan biopolymer and other application of the chitosan in food packaging (i.e. coatings for paper/board, composite films with mineral nanoparticles – ZnO…etc) knowing the fact that the in these type of packaging chitosan is used as biopolymer matrix to embed other compounds with active and functional properties.

Response 5: Further information on this issue was inserted on Lines 246-248 and 257-261.

  1. In the conclusion section it is mentioned also,.. ”From another perspective, the conventional extraction of both biomolecules requires the use of hazard chemicals such as sodium hydroxide and hydrochloric acid. However, innovative technological approaches, as the use of green solvents, can once again be used to replace hazard chemicals, making the extraction process more efficient and more environmentally friendly…” the using of green solvent to extract the chitin from seafood waste, but in the manuscript is not described these extraction methods…

Response 6: The use of green solvents to chitin/chitosan extraction was presented on Lines 138-150.

Reviewer 2 Report

Dear Authors,

Your manuscript is well structured and organized. Several comments are given in the attached file. Please, address these comments and your review article will be much better.

Thanks.  

Author Response

Dear Reviewer,

The authors appreciated your careful evaluation and comments. We have incorporated all your suggestions and made the appropriate corrections. Please, observe that we have incorporated two figures in the text to improve the manuscript comprehension. All changes made were written in red in the revised manuscript.

Reviewer comments

  1. can you find newer data? e.g. in last 5 years?

Response 1: Recent studies by Kruijssen et al. [15], Maulu et al. [20], Tacias-Pascacio et al. [21] were inserted in the text (lines 88-95).

2.  again, newer data will suit better to this review article.

Response 2: Data from a recent report from FAO [19] was insert in the text (lines 69-74) and a Figure showing the major types of marine crustaceans globally produced by capture over the last two decades was inserted (Figure 1).

3. Newer data about chitosan market.

Response 3: Recent data about the Chitosan market was insert in the text (lines 117-123). 

4. G-capital letter R-   ...A...S...

Response 4: The sentence was amended (lines 216).

5. Reference?

Response 5: References [34, 47] were inserted in line 232.

6. most of the references in this part are not suitable. e.g. [67,68] is about active packaging and not about different materials for food pack...

Response 6: More apropriate references were inserted in the text (lines 275-281).

7. you can add more novel references on topic: edible packaging..Ref. from 2013 is relatively old reference

Response 7: More recent references were inserted in the text (line 310).